# Structural insights into the Ca$^{2+}$-dependent gating of the human mitochondrial calcium uniporter

Yan Wang[1,2,3], Yan Han[1,2,3], Ji She[1,2†], Nam X Nguyen[1,2,3‡], Vamsi K Mootha[4], Xiao-chen Bai[2,5], Youxing Jiang[1,2,3]*

[1]Department of Physiology, University of Texas Southwestern Medical Center, Dallas, United States; [2]Department of Biophysics, University of Texas Southwestern Medical Center, Dallas, United States; [3]Howard Hughes Medical Institute, University of Texas Southwestern Medical Center, Dallas, United States; [4]Howard Hughes Medical Institute and Department of Molecular Biology, Massachusetts General Hospital, Harvard Medical School, Broad Institute, Cambridge, United States; [5]Department of Cell Biology, University of Texas Southwestern Medical Center, Dallas, United States

*For correspondence:
youxing.jiang@utsouthwestern.
edu

Present address: †University of
Science and Technology of
China, Hefei, China; ‡Regeneron
Pharmaceuticals, Inc, Tarrytown,
United States

Competing interests: The
authors declare that no
competing interests exist.

Reviewing editor: Kenton J
Swartz, National Institute of
Neurological Disorders and
Stroke, National Institutes of
Health, United States

**Abstract** Mitochondrial Ca$^{2+}$ uptake is mediated by an inner mitochondrial membrane protein called the mitochondrial calcium uniporter. In humans, the uniporter functions as a holocomplex consisting of MCU, EMRE, MICU1 and MICU2, among which MCU and EMRE form a subcomplex and function as the conductive channel while MICU1 and MICU2 are EF-hand proteins that regulate the channel activity in a Ca$^{2+}$-dependent manner. Here, we present the EM structures of the human mitochondrial calcium uniporter holocomplex (uniplex) in the presence and absence of Ca$^{2+}$, revealing distinct Ca$^{2+}$ dependent assembly of the uniplex. Our structural observations suggest that Ca$^{2+}$ changes the dimerization interaction between MICU1 and MICU2, which in turn determines how the MICU1-MICU2 subcomplex interacts with the MCU-EMRE channel and, consequently, changes the distribution of the uniplex assemblies between the blocked and unblocked states.

## Introduction

Intracellular Ca$^{2+}$ signaling can mediate an array of biological processes ranging from transcriptional and metabolic regulation to cell death (*Berridge et al., 2003*). It has long been observed that the vast mitochondrial network takes up large amounts of Ca$^{2+}$ from its environment and buffers cytosolic Ca$^{2+}$ elevations, thus regulating the spatial and temporal dynamics of intracellular Ca$^{2+}$ signaling (*Clapham, 2007*; *Kamer and Mootha, 2015*; *Rizzuto et al., 2012*). Mitochondrial calcium uptake is mediated by the mitochondrial calcium uniporter, a Ca$^{2+}$-selective channel that is localized to the inner mitochondrial membrane (*Gunter and Pfeiffer, 1990*; *Kirichok et al., 2004*). In humans, the uniporter functions as a holocomplex (referred to as uniplex henceforth) and consists of the pore-forming subunit MCU (*Baughman et al., 2011*; *De Stefani et al., 2011*), an essential membrane-spanning subunit EMRE (*Sancak et al., 2013*), and the gate-keeping subunits MICU1 (*Perocchi et al., 2010*) and MICU2 (*Plovanich et al., 2013*). The two-transmembrane (2-TM) MCU component forms a tetrameric channel pore and the single-TM EMRE becomes an integral part of the channel by forming a 1:1 stoichiometric subcomplex with MCU and renders the channel conductive (*Wang et al., 2019*). Thus, an MCU-EMRE subcomplex constitutes the minimal component necessary and sufficient for mitochondrial calcium uptake (*Kovács-Bogdán et al., 2014*; *Sancak et al., 2013*). MICU1 and MICU2 are two paralogous EF-hand containing proteins that are localized to the

mitochondrial intermembrane space (IMS) and function as gate-keepers by sensing the cytosolic $Ca^{2+}$ concentration and, thereby, regulate MCU-EMRE channel activity (*Csordás et al., 2013*; *Kamer et al., 2017*; *Mallilankaraman et al., 2012*). Each MICU subunit contains four EF-hand motifs among which only EF-1 and EF-4 are canonical and bind $Ca^{2+}$, whereas EF-2 and EF-3 lack the essential residues for $Ca^{2+}$ binding.

How MICU1 and MICU2 interact with MCU and exert their $Ca^{2+}$-dependent gating properties on the uniporter has been a central question with regard to the function and regulation of the uniplex. Functional and biochemical analyses seem to converge on the following observations: MICU1 and MICU2 form heterodimers with a disulfide bond between them (*Kamer and Mootha, 2014*; *Patron et al., 2014*); MICU1 but not MICU2 interacts with the MCU pore (*Kamer and Mootha, 2014*; *Plovanich et al., 2013*), which is mediated by EMRE through electrostatic interactions between positively charged residues (KKKKR poly-basic region) at the N-terminus of MICU1 and the C-terminal poly-aspartate tail of EMRE (*Tsai et al., 2016*); recent studies also suggested that MICU1 makes direct interaction with the aspartate ring (D-ring) of MCU formed by the canonical DIME motif in the selectivity filter of an MCU channel tetramer (*Paillard et al., 2018*; *Phillips et al., 2019*). Thus, the current model of MCU gating posits that at low cytosolic $Ca^{2+}$ concentrations, MICU1 functions as a pore blocker and prevents $Ca^{2+}$ uptake into the mitochondrial matrix and that MCU becomes disinhibited only when cytosolic $Ca^{2+}$ approaches the micromolar range, resulting from cooperative $Ca^{2+}$ binding to the EF-hands of MICU1 and MICU2 (*Csordás et al., 2013*; *Kamer et al., 2017*; *Mallilankaraman et al., 2012*). Intriguingly, the deleterious effects of aberrant mitochondrial calcium signaling are manifested only in patients and animal models with defects in MICU1 rather than MCU (*Bick et al., 2017*; *Liu et al., 2016*; *Logan et al., 2014*; *Musa et al., 2019*), thus highlighting the impetus to pursue a better mechanistic understanding of the gating properties of the uniplex.

Since the molecular identification of the uniplex composition, multiple structures of individual components or subcomplexes have been determined, including several fungal MCU orthologs (*Baradaran et al., 2018*; *Fan et al., 2018*; *Nguyen et al., 2018*; *Yoo et al., 2018*), the human MCU-EMRE subcomplex (*Wang et al., 2019*) and the mammalian MICU1 and MICU2 proteins in the homo- and heterodimeric forms (*Kamer et al., 2019*; *Park et al., 2020*; *Wang et al., 2014*; *Wu et al., 2019*; *Xing et al., 2019*). However, to understand the molecular mechanisms of the uniplex assembly and its $Ca^{2+}$-dependent gating necessitates the structural determination of the four-component MCU holocomplex. To this end, several groups have recently reported the structures of the MCU holocomplex in various states, including the structures of the human MCU holocomplex in low and high $[Ca^{2+}]$ (*Fan et al., 2020*), the structure of human MCU holocomplex in the absence of $Ca^{2+}$ (*Zhuo et al., 2020*), and the structure of a holocomplex formed by the beetle MCU-EMRE and the human MICU1-MICU2 under resting $[Ca^{2+}]$ conditions (*Wang et al., 2020*). Here, we also present the cryo-EM structures of the human MCU holocomplex in the apo and $Ca^{2+}$-bound states. Our structural study reveals multiple, distinct uniplex assemblies that likely represent the different functional states of the uniporter and provides structural insights into the $Ca^{2+}$-dependent gate-keeping function of MICU1 and MICU2.

## Results

### Biochemistry and structure determination of the human MCU holocomplex

Our previous structural study demonstrated that human MCU and EMRE together function as a conductive channel by assembling into a tightly packed subcomplex in which four MCU subunits form the ion conduction pore, which is stabilized by four EMRE subunits that sits on the periphery (*Wang et al., 2019*; *Figure 1—figure supplement 1*). Two MCU-EMRE subcomplexes can further dimerize through electrostatic interactions along an interface at the N-terminal domain of MCU. To obtain the uniporter holocomplex (or uniplex), we first prepared the MCU-EMRE subcomplex by co-expressing MCU and EMRE in HEK293 cells and purifying the subcomplex in detergent followed by reconstitution into lipid nanodisc as previously described (*Wang et al., 2019*). MICU1 and MICU2 were expressed individually in *E. coli* and purified as described in the Materials and methods. The uniplex consisting of MCU-EMRE-MICU1-MICU2 was then reconstituted in vitro by mixing the MCU-EMRE subcomplex nanodisc with excessive amounts of MICU1 and MICU2 followed by purification

using size exclusion chromatography (SEC). To obtain the uniplex samples in the Ca²⁺-bound and Ca²⁺-free state for cryo-EM studies, the buffers used in the final SEC purification step contained either 2 mM Ca²⁺ or 2 mM EGTA without Ca²⁺, respectively. The SEC elution volume of the purified uniplex indicates a size much larger than the dimeric form of the MCU-EMRE subcomplex, suggesting that the uniplex also forms a dimer in solution (*Figure 1—figure supplement 2*).

3D classification of the uniplex sample prepared in the presence of Ca²⁺ yielded one major class that shows clear density of a V-shaped dimer of the MCU-EMRE subcomplexes with their tops bridged by a heterotetramer of MICU1 and MICU2 (*Figure 1a* and *Figure 1—figure supplement 3*). To better resolve the MICU1 and MICU2 in the uniplex, we subsequently performed focused 3D classification by masking around the region of the MICU1-MICU2 heterotetramer, and the final EM map was refined to 4.2 Å (Materials and methods, *Figure 1—figure supplement 3* and *Figure 1—source data 1*). On the contrary, the particles of the uniplex sample prepared in the absence of Ca²⁺ were heterogeneous and could be classified into four classes with distinct conformations, indicating a diverse spectrum of uniplex assemblies (*Figure 1b*, *Figure 1—figure supplement 4*, and *Figure 1—source data 1*). The major class of particles (class 1) represents a blocked uniplex in which a heterodimer of MICU1 and MICU2 blocks the central ion conduction pore of a monomeric MCU-EMRE subcomplex. Similar MICU1-MICU2 blocking is also observed in the second class (class 2) with the exception being that MCU-EMRE subcomplex retains its dimeric form. By C2 symmetry

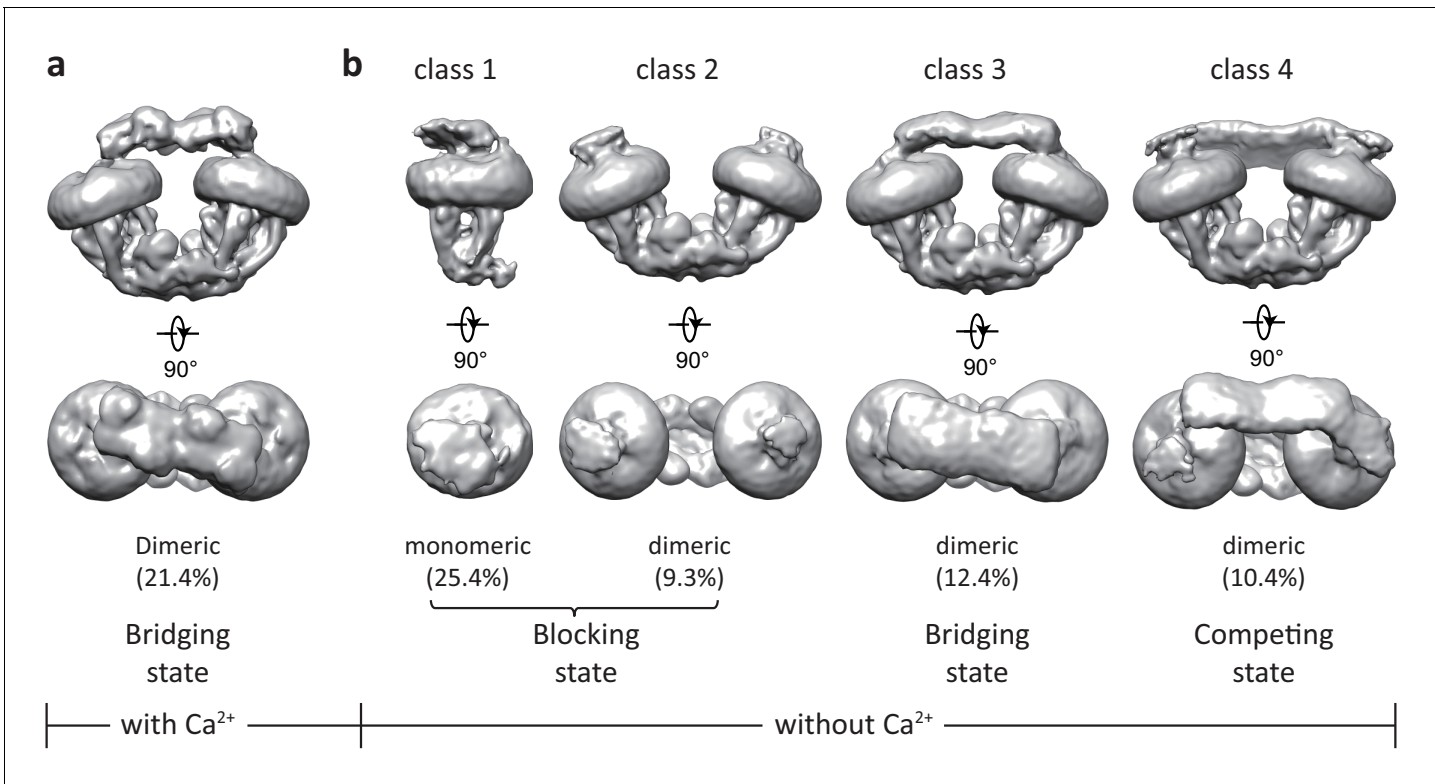

**Figure 1.** Initial 3D classifications of the human MCU-EMRE-MICU1-MICU2 holocomplex (uniplex) in the presence and absence of Ca²⁺. Numbers in parentheses denote the percentage of particles for each class. (**a**) The major class of particles from the uniplex sample prepared in the presence of 2 mM Ca²⁺. (**b**) Four classes of particles from the uniplex sample prepared in the absence of Ca²⁺ (with 2 mM EGTA).

The online version of this article includes the following source data and figure supplement(s) for figure 1:

**Source data 1.** Cryo-EM data collection, refinement and validation statistics.

**Figure supplement 1.** Structure of the MCU-EMRE subcomplex.

**Figure supplement 2.** Purification of the human MCU-EMRE-MICU1-MICU2 holocomplex reconstituted into nanodisc.

**Figure supplement 3.** Cryo-EM data processing scheme of the uniplex assembly in the presence of Ca²⁺.

**Figure supplement 4.** Cryo-EM data processing scheme of the uniplex assemblies in the absence of Ca²⁺.

**Figure supplement 5.** Cryo-EM data processing scheme of the apo blocked uniplex.

**Figure supplement 6.** Sequence and secondary structure assignment of human MICU1 (upper) and MICU2 (lower).

expansion, particles of this class were combined with those of class one and yielded the structure of a blocked uniplex at 4.6 Å resolution (Materials and methods, *Figure 1—figure supplements 4* and *5*, and *Figure 1—source data 1*). The MICU1 and MICU2 in the EM maps of classes 3 and 4 were poorly resolved relative to the rest of the uniplex, likely due to their conformational heterogeneity. In the EM-map of class three that was determined at 4.7 Å, MICU1 and MICU2 form a heterotetramer and bridge the dimeric form of the MCU-EMRE subcomplex similar to that observed in the $Ca^{2+}$-bound uniplex. Interestingly, the structure of class four represents a competing conformation in which both the MICU1-MICU2 heterotetramer – as a bridge – and the MICU1-MICU2 heterodimers – as pore blockers – are bound to the same MCU-EMRE subcomplex dimer. Because of this competition, the bridging MICU1-MICU2 tetramer is tilted to the side of the MCU-EMRE subcomplex. The particles of this class were highly heterogeneous and we were only able to reconstruct a 7 Å map with definable density for MICU1 and MICU2.

In all these structures, the MCU-EMRE subcomplex appears to be the most stable part of the uniplex, as indicated by the well-defined density, whereas MICU1 and MICU2 are more dynamic and engage in the uniplex assembly with different configurations depending on the presence or absence of $Ca^{2+}$. Despite their low overall resolutions, the EM density for the MCU-EMRE subcomplex is well defined in all of the maps and can be perfectly modeled by rigid-body docking of the previously determined structure with minor adjustments (*Wang et al., 2019*). In addition, multiple crystal structures of human MICU1 and MICU2 homodimer in both the apo and $Ca^{2+}$-bound states, as well as the apo MICU1-MICU2 heterodimer have been determined (*Kamer et al., 2019*; *Park et al., 2020*; *Wang et al., 2014*; *Wu et al., 2019*; *Xing et al., 2019*), thus facilitating our model building for the MICU1 and MICU2 part of the uniplex (*Figure 1—figure supplement 6*). As the structures of the individual MICU1, MICU2 and MCU-EMRE subcomplex have all been extensively described before, our discussion will be focused on how MICU1 and MICU2 interact with MCU-EMRE to form a uniplex assembly in a $Ca^{2+}$-dependent manner.

## Uniplex assembly in the presence of $Ca^{2+}$

In the presence of $Ca^{2+}$, the uniplex adopts a two-fold symmetry and consists of a V-shaped dimer of MCU-EMRE subcomplexes and a MICU1-MICU2 heterotetramer that bridges the tops of the two subcomplexes (*Figure 2a&b*). The MICU1-MICU2 tetramer is formed by the dimerization of two MICU1-MICU2 heterodimers through a back-to-back packing between MICU2 subunits (*Xing et al., 2019*), resulting in a linear arrangement of MICU1-MICU2-MICU2-MICU1. MICU1 and MICU2 dimerize through a face-to-face contact in which the EF-1 motif of one subunit interacts with the EF-3 of the other (*Figure 2c*). This face-to-face dimerization mode has been commonly observed in the crystal structures of MICU1 and MICU2 homodimers as well as the MICU1-MICU2 heterodimer (*Park et al., 2020*; *Wang et al., 2014*; *Wu et al., 2019*; *Xing et al., 2019*). Major $Ca^{2+}$-induced conformational changes also occur at the interface between MICU1 and MICU2 as will be further discussed later.

The assembly of the MICU1-MICU2 tetramer and MCU-EMRE subcomplexes in the presence of $Ca^{2+}$ is mediated by the interactions between MICU1 and EMRE. Each MICU1 subunit interacts with two EMRE subunits that are proximal to the central axis of the uniplex dimer by forming putative salt-bridge networks (*Figure 2d–f*). Firstly, the α1 helix of MICU1 is oriented parallel to the membrane surface with its N-terminus pointing toward the C-terminal end of the transmembrane helix of EMRE (*Figure 2e*). This configuration would lead to the convergence of the KKKKR poly-basic region that extends from the MICU1 α1 helix to the C-terminal poly-aspartate tail on EMRE. Although these two charged regions are not resolved in the structure, their close proximity would imply the formation of a network of salt bridges between them. Both charged regions are highly conserved and have been shown to be important for uniplex assembly and $Ca^{2+}$ uptake (*Tsai et al., 2016*). The second point of contact is mediated by the α11 helix of MICU1 and the poly-aspartate tail of the other EMRE subunit. At this site, the poly-aspartate tail of EMRE forms an extended loop that runs parallel to and is positioned below the α11 helix of MICU1 (*Figure 2f*). The MICU1 α11 helix contains multiple basic residues whose side chains point directly toward the EMRE loop and likely form the second network of salt bridges with the poly-aspartate tail of EMRE. The binding of the MICU1-MICU2 tetramer does not block the channel pore of MCU, nor does it introduce any visible conformational change to each MCU-EMRE subcomplex, whose structure is virtually identical to that determined previously without bound MICU1 and MICU2 (*Figure 2—figure supplement 1a*). Thus, the uniplex

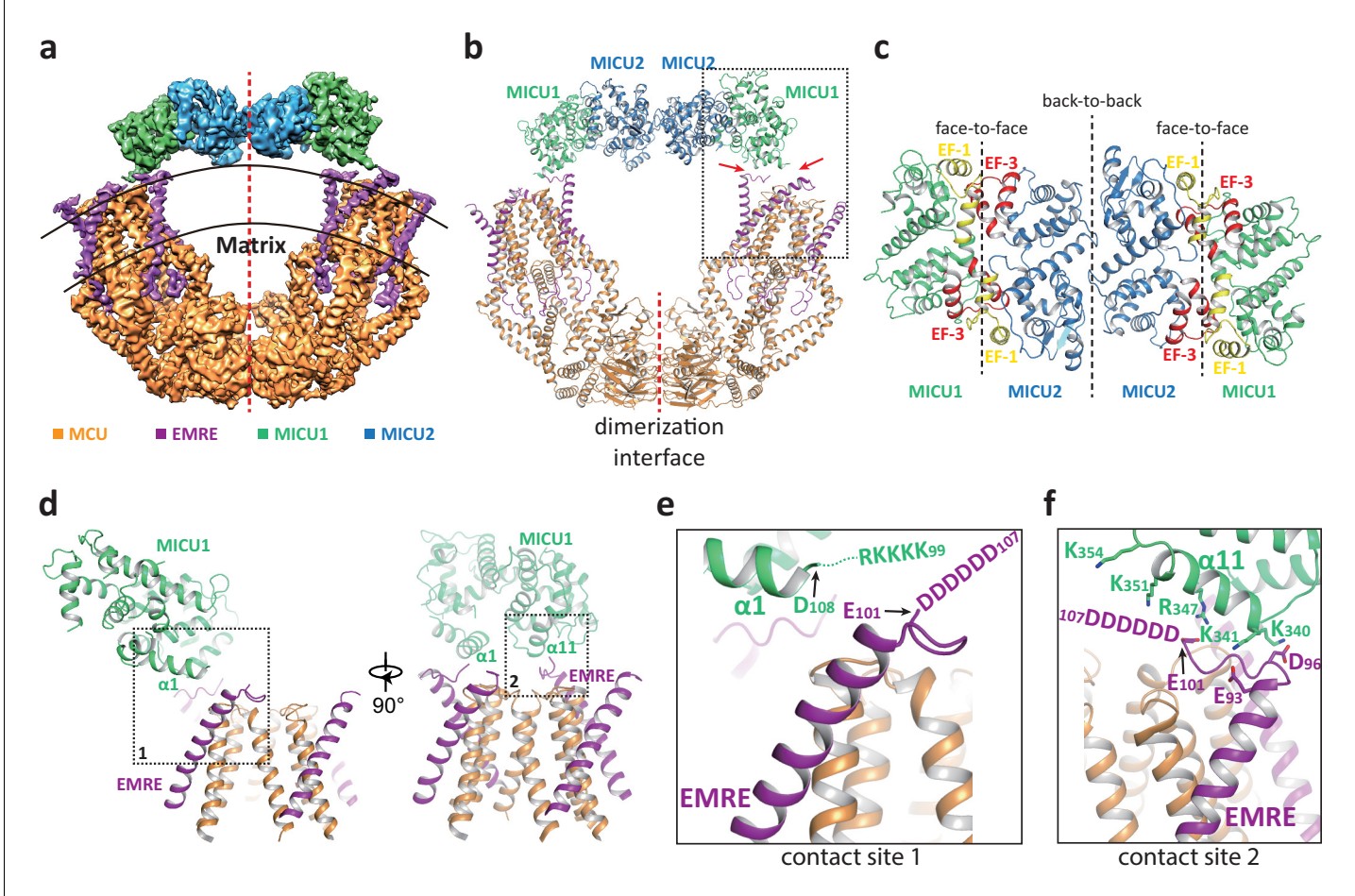

**Figure 2.** Structure of the MCU-EMRE-MICU1-MICU2 uniplex assembly in the presence $Ca^{2+}$. (**a**) Side view of the 3D reconstruction of the $Ca^{2+}$-bound uniplex with each protein component individually colored. Red line marks the two-fold axis of the complex. (**b**) Cartoon representation of the $Ca^{2+}$-bound uniplex structure. Red arrows mark the two contact sites between MICU1 and MCU/EMRE. Red line marks the dimerization interface between two MCU/EMRE subcomplexes. (**c**) Down view of the MICU1-MICU2 heterotetramer in cartoon representation along the two-fold axis. EF-1 and EF-3 motifs are colored in yellow and red, respectively. Dotted lines mark the interfaces between neighboring MICU subunits. (**d**) Protein-protein interface between the MICU1 and MCU/EMRE subcomplex (boxed area in (**b**)) with the two contact sites numbered and boxed. (**e**) Zoomed-in view of the contact site 1. In the $Ca^{2+}$-bound uniplex, the structure model for MICU1 starts at D108 and the model for EMRE ends at E101. (**f**) Zoomed-in view of the contact site 2.

The online version of this article includes the following figure supplement(s) for figure 2:

**Figure supplement 1.** Structural comparison between the MCU-EMRE subcomplexes with and without bound MICU1-MICU2.

structure obtained in the presence of $Ca^{2+}$ represents the unblocked state in which the channel pore is poised for $Ca^{2+}$ uptake. However, the tethering of two MCU-EMRE subcomplexes by the MICU1-MICU2 tetramer appears to exert a pulling force on the MCU-EMRE subcomplexes, resulting in a slightly narrower V-shaped dimer as compared to the subcomplex dimer structure previously determined without MICU1-MICU2 (*Figure 2—figure supplement 1b*).

## Uniplex assembly with a blocked channel pore in the absence of $Ca^{2+}$

Multiple uniplex assemblies were observed in the absence of $Ca^{2+}$, among which the blocking of the central channel pore by a MICU1-MICU2 heterodimer is the dominant form. This MICU1-MICU2 blocking event occurs on both the monomeric and dimeric forms of the MCU-EMRE subcomplex. As the MCU-EMRE subcomplex remained dimeric in the purified uniplex sample, we suspect that the particles of blocked monomeric subcomplex likely come from the dissociation of the subcomplex dimer during EM grid preparation. For simplicity, the monomeric form of the subcomplex will be

used in the following discussion. With four-fold symmetry at the channel pore of MCU-EMRE, the MICU1-MICU2 dimer can bind and block the channel in four possible orientations (*Figure 3—figure supplement 1*). Symmetry expansion was used to align all the MICU1-MICU2 dimers in the same orientation so that they could be averaged together in the 3D reconstruction, yielding an EM map with clear density for the MICU1-MICU2 dimer and the symmetrical part of the MCU-EMRE subcomplex, which includes EMRE and the transmembrane domain (TMD) and the coiled coil domain (CCD) of MCU (*Figure 3a* and *Figure 1—figure supplement 5*). The density for the non-symmetrical linker-helix domain (LHD) and N-terminal domain (NTD) of MCU was averaged out.

In the blocked uniplex structure, MICU1 engages with the MCU-EMRE subcomplex at three major points of contact using its N- and C-terminal helices (*Figure 3b*). Firstly, the α1 helix of MICU1 is oriented directly above the ion conduction pore with its N-terminus pointing toward the C-terminus of the EMRE helix. This would position the N-terminal poly-basic region of MICU1 within the proximity of the C-terminal poly-aspartate tail of EMRE, allowing for the formation of a putative salt-bridge network (*Figure 3c*). Although similar salt bridge interactions are also proposed in the Ca$^{2+}$-bound uniplex structure, the orientation of the α1 helix is quite the opposite between the two structures. In the blocked uniplex, the C-terminus of the MICU1 α1 helix points toward the center of the channel rather than away from the channel as seen in the Ca$^{2+}$-bound structure. Secondly, the α2 helix is positioned directly above the D-ring of MCU's selectivity filter, which is formed by the four Asp261s from the DIME motif at the external entrance of the channel tetramer (*Figure 3d*). Several basic residues from MICU1 are in close proximity and can potentially engage in electrostatic interactions with the MCU D-ring Asp261s, including K126 and R129 in α2 as well as the basic resides (R259, R261 and R263) in the loop between α7 and α8, which is disordered in the Ca$^{2+}$-bound MICU1. Most notably, Lys126 is well positioned to point its side chain directly toward the center of the D-ring and block the channel filter. The same electrostatic interactions have been observed in the recent structure of the human uniplex obtained in low [Ca$^{2+}$] (*Fan et al., 2020*) as well as the structure of a holo-complex formed by the beetle MCU-EMRE and the human MICU1-MICU2 under resting [Ca$^{2+}$] conditions (*Wang et al., 2020*). In both studies, it has also been shown that mutations of these positively charged residues in MICU1 destabilize the MCU-MICU1 interactions and mitigate the blocking of MCU by MICU1. Thirdly, the amphipathic C-terminal helix (C-helix, α17) of MICU1 lies on the membrane surface between two neighboring EMRE subunits with its N-terminus sandwiched between the C-termini of an EMRE helix and the TM1 from a neighboring MCU subunit (*Figure 3e*). Although the protein side chains are not resolved in the structure, the close proximity of multiple hydrophobic residues at this contact site would imply that van der Waals contacts mediate the interactions between MICU1's C-helix and MCU-EMRE. Deletion of MICU1's C-helix has been shown to weaken the interaction between MCU and MICU1 (*Fan et al., 2020*; *Wang et al., 2014*).

## Unblocked uniplex assembly in the absence of Ca$^{2+}$

While the blocked uniporter represents the majority of the uniplex particles in the absence of Ca$^{2+}$, a smaller class of particles reveals the unblocked uniporter with the MICU1-MICU2 heterotetramer bridging the two MCU-EMRE subcomplexes similar to that observed in the presence of high Ca$^{2+}$ (*Figure 4a*). In this apo, unblocked uniplex, the MCU-EMRE subcomplex part of the structure can be resolved to 3.7 Å (*Figure 1—figure supplement 4*). However, the MICU1 and MICU2 parts of the uniplex are poorly resolved in the EM map, suggesting that they are highly mobile and loosely attached to the MCU-EMRE subcomplex. Intriguingly, the MICU1-MICU2 tetramer in the apo uniplex has an overall structural arrangement as that of the Ca$^{2+}$-bound uniplex, but it does not appear to induce any movement to the MCU-EMRE subcomplexes whose V-shaped dimer structure remains identical to that determined without MICU1 and MICU2 (*Figure 2—figure supplement 1c*). Interestingly, similar apo, bridged uniplex structure is also observed in a recent study of the human uniplex obtained in the absence of Ca$^{2+}$ (*Zhuo et al., 2020*). The high degree of mobility exhibited by the MICU1-MICU2 tetramer in the apo, unblocked uniplex is also evident in another class of particles in which the bridging MICU1-MICU2 tetramer is pushed to the side of the MCU-EMRE subcomplex due to the competing binding of the MICU1-MICU2 dimer that occludes the channel pore (*Figure 4b*). The structures reconstructed from both classes of particles are of low resolution and do not allow us to define the interactions between MICU1 and the MCU-EMRE subcomplex. Nevertheless, the observation that the MICU1-MICU2 tetramer in a smaller subset of the apo uniplex particles

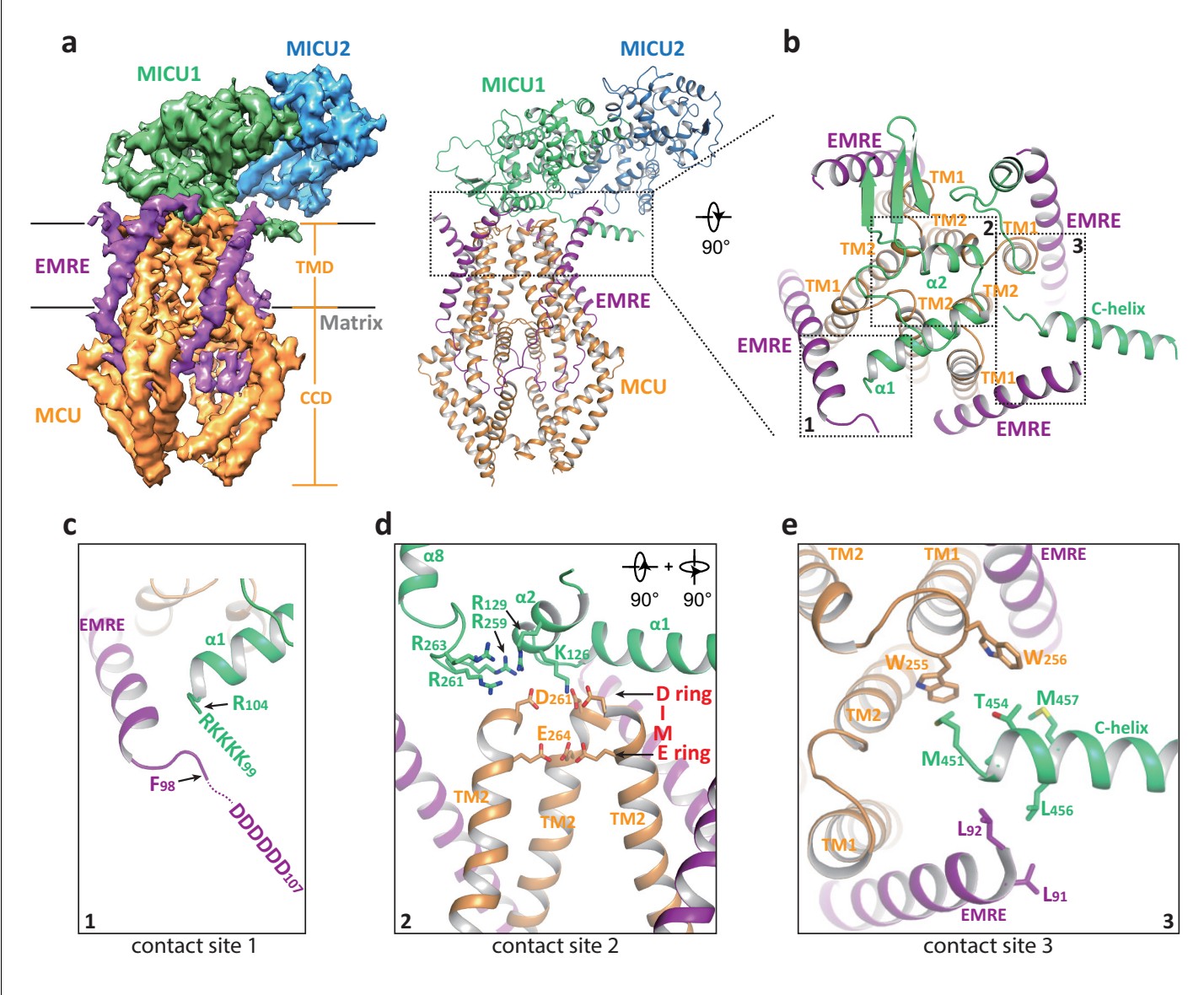

**Figure 3.** Structure of the apo, blocked MCU-EMRE-MICU1-MICU2 uniplex assembly. (**a**) Side views of the 3D reconstruction (left) and cartoon representation (right) of the apo, blocked uniplex with each protein component individually colored. The map and model contain only the transmembrane domain (TMD) and the coiled coil domain (CCD) of MCU. The interface between MICU1 and MCU-EMRE subcomplex is boxed. (**b**) Zoomed-in view of the interface between MICU1 and the MCU/EMRE subcomplex with the three contact sites numbered and boxed. (**c**) Interactions between MICU1's poly-basic sequence (KKKKR) and EMRE's C-terminal poly-Asp tail. In the apo, blocked uniplex, the structure model for MICU1 starts at R104 and the model for EMRE ends at F98. (**d**) Interactions between MICU1 and MCU's D-ring at the selectivity filter. MCU's front subunit is removed for clarity. (**e**) Interactions between MICU1's C-helix and MCU/EMRE.

The online version of this article includes the following figure supplement(s) for figure 3:

**Figure supplement 1.** MICU1-MICU2 blocking of the MCU-EMRE pore with four possible orientations.

and its weak association with the MCU-EMRE subcomplexes implies that the MICU1-MICU2 tetramer is not stable in the absence of $Ca^{2+}$ and does not form tight contacts with the channel.

## $Ca^{2+}$-dependent conformational change at MICU1-MICU2

Comparison of uniplex structures with or without $Ca^{2+}$ reveals that the $Ca^{2+}$-induced conformational changes mainly occur within the MICU1-MICU2 dimer. The structural changes within individual MICU1 or MICU2 subunit are quite subtle with the most noticeable difference observed at the EF-1

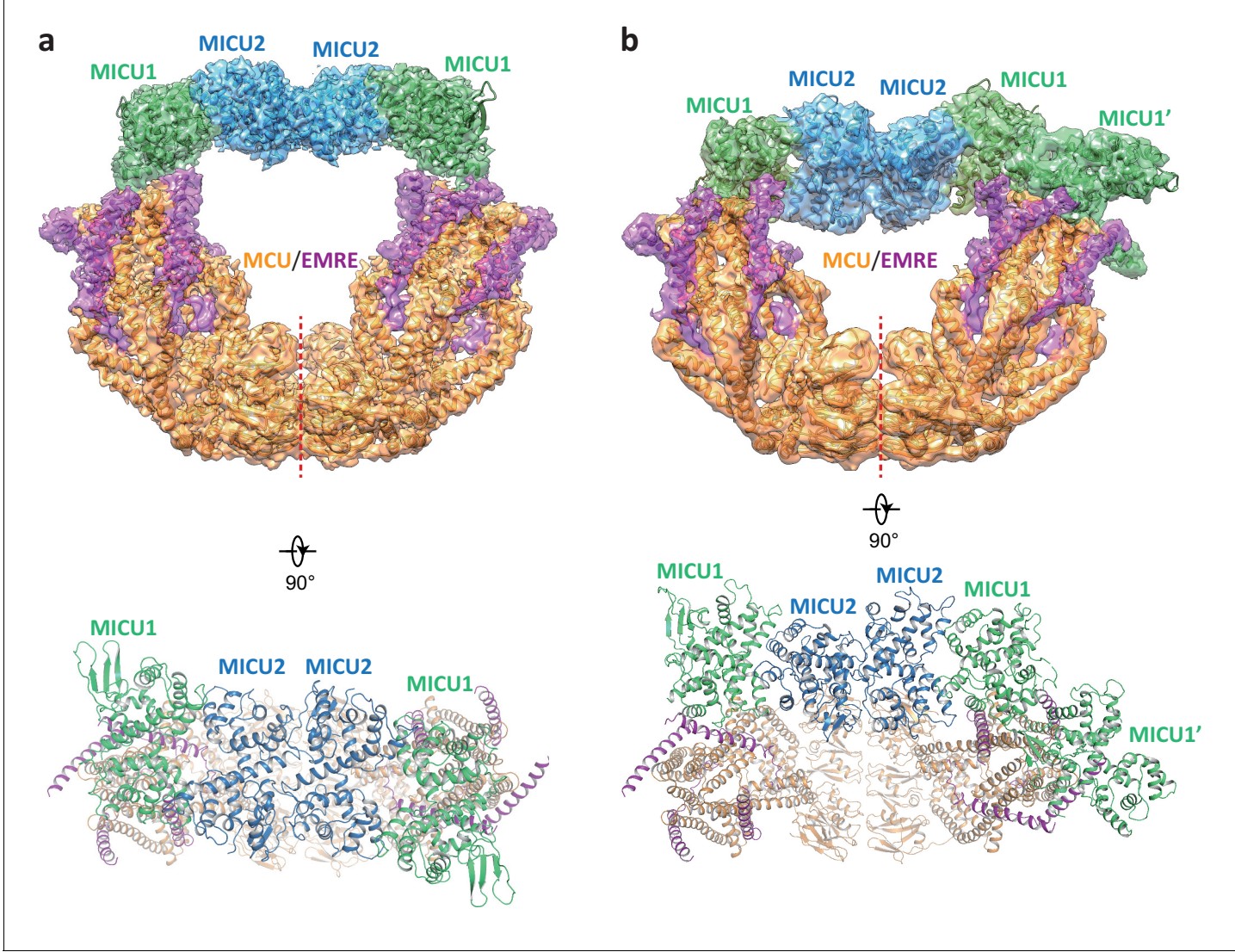

**Figure 4.** Structures of the apo, bridged MCU-EMRE-MICU1-MICU2 uniplex assembly. (**a**) 3D reconstruction (upper, side view) and cartoon representation (lower, top view) of the apo, bridged uniplex with each protein component individually colored. Red line marks the dimerization interface between two MCU/EMRE subcomplexes. (**b**) 3D reconstruction (upper, side view) and cartoon representation (lower, top view) of the apo, bridged uniplex with competing binding of MICU1-MICU2 that blocks the pore. We were only able to model the MICU1 subunit (labeled as MICU1') of the MICU1-MICU2 dimer that blocks the MCU-EMRE subcomplex on the right side. Note that the bridging MICU1-MICU2 tetramer is tilted to the side of the MCU-EMRE subcomplexes.

motif where the two helices ($\alpha6$ as the E1 helix and $\alpha7$ as the F1 helix in both MICU1 and MICU2) are relatively parallel in the apo state but swing apart and become more perpendicular to each other in the $Ca^{2+}$-bound state (*Figure 5a*). As the EF-1 motifs from both MICU1 and MICU2 are directly involved in their face-to-face dimerization by forming hydrophobic interactions with the EF-3 motifs ($\alpha13$ as the E3 helix and $\alpha14$ as the F3 helix in both MICU1 and MICU2) from their respective partners, the conformational change at EF-1 results in a rearrangement of the hydrophobic contacts at the dimerization interface. Consequently, MICU1 and MICU2 form a more compact dimer in the $Ca^{2+}$-bound state and their relative position also differs from that of the apo dimer (*Figure 5b*).

The change in the relative position between MICU1 and MICU2 provides a plausible structural explanation for the destabilization of channel blocking by MICU1-MICU2 upon $Ca^{2+}$ binding. If we allow the $Ca^{2+}$-bound MICU1-MICU2 dimer to make the same blocking interactions with MCU-EMRE as the apo dimer – by superimposing its MICU1 subunit onto the MICU1 of the apo blocked

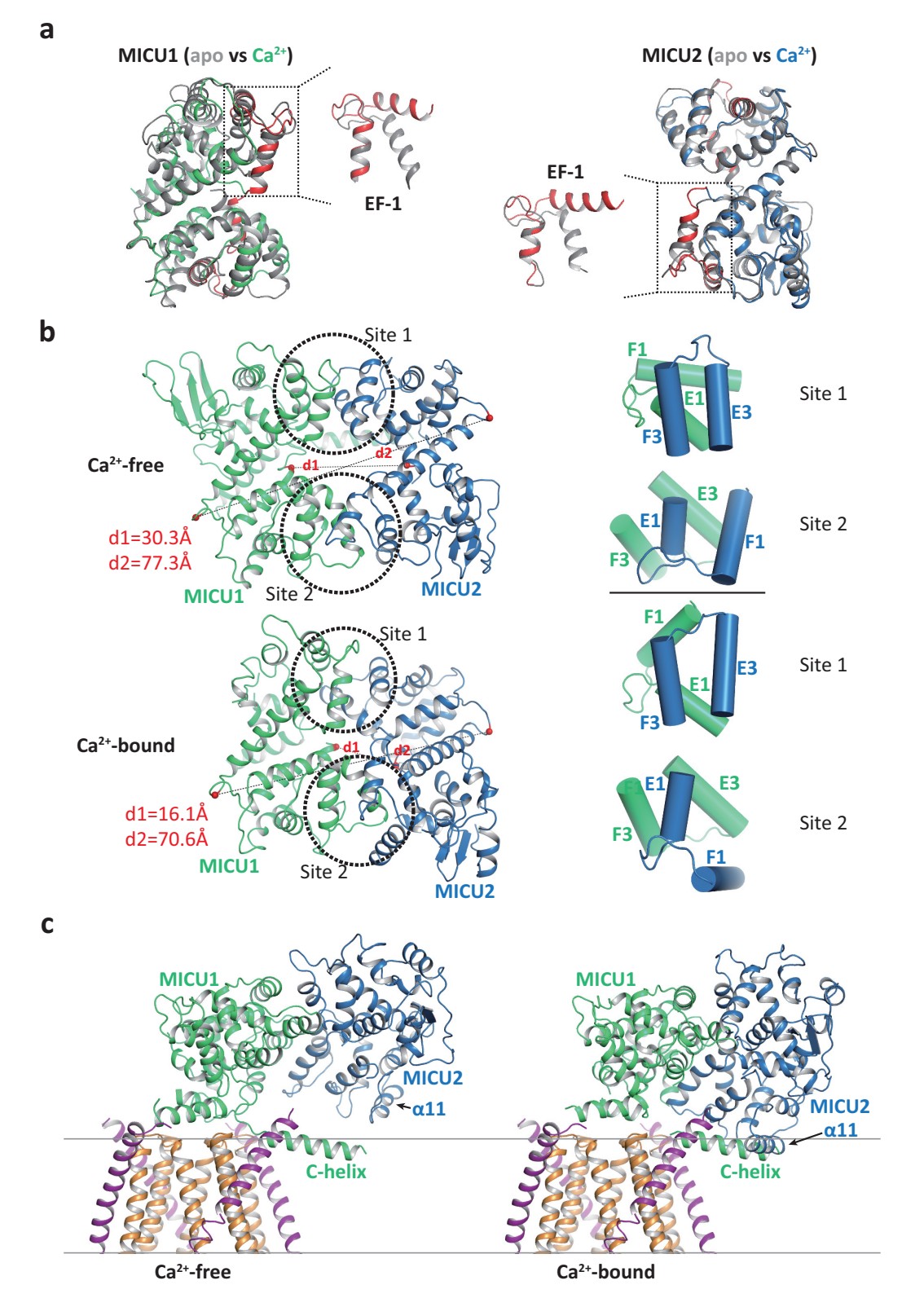

**Figure 5.** Structural comparison of the MICU1-MICU2 dimers in the apo and Ca$^{2+}$-bound states. (**a**) Superimposition of individual MICU1 (left) or MICU2 (right) subunit structures between the apo (grey) and the Ca$^{2+}$-bound (MICU1 in green and MICU2 in blue) states. EF-1 and EF-3 motifs of MICU1 and MICU2 in Ca$^{2+}$-bound states are colored red. (**b**) Structural comparison of the MICU1-MICU2 heterodimer in the apo (upper) and Ca$^{2+}$-bound (lower) states with the inter-subunit contact sites circled. Grey dashed lines mark the inter-subunit distances between the Ca atoms (red spheres) of MICU1's

*Figure 5 continued on next page*

*Figure 5 continued*
Met442 and MICU2's His396 (d1) and between the Ca atoms of MICU1's Val318 and MICU2's Gly274 (d2). Right panels show the inter-subunit packing between EF-1 and EF-3 at each contact site in the absence (upper two) and presence (lower two) of Ca$^{2+}$. (c) Left, structure of the blocked MCU-EMRE channel pore by a MICU1-MICU2 dimer in the apo state. Right, hypothetical model of the channel pore being blocked by the Ca$^{2+}$-bound MICU1-MICU2 dimer. The model is generated by superimposing the MICU1 subunit of the Ca$^{2+}$-bound MICU1-MICU2 dimer onto the MICU1 of the apo, blocked uniplex.

uniplex – then the MICU2 subunit would be positioned much closer to the membrane surface and its α11 helix would directly clash with the C-helix of MICU1 or the membrane surface (*Figure 5c*). This steric interference implies that the Ca$^{2+}$-induced change at the interfacial contact between MICU1 and MICU2, and the ensuing movement of MICU2 relative to MICU1, would destabilize the blocking interaction between MICU1 and MCU-EMRE and thus prevent the MICU1-MICU2 dimer from blocking the channel.

## Discussion

Here, we present multiple structures of the human uniplex consisting of MCU-EMRE-MICU1-MICU2 in the presence and absence of Ca$^{2+}$. Consistent with our previous study (*Wang et al., 2019*), the MCU-EMRE subcomplex forms a dimer in the uniplex assembly, which likely represents a physiologically relevant state of the uniporter. MICU1-MICU2 appears to be more stable as a heterodimer in the absence of Ca$^{2+}$, but can further dimerize to form a heterotetramer upon Ca$^{2+}$ binding through back-to-back dimerization between two MICU2 subunits. As no obvious structural change is observed at the MICU2-MICU2 homo-dimerization interface between the apo and Ca$^{2+}$-bound states, it is unclear how Ca$^{2+}$ binding promotes the tetramer formation. One possibility is that Ca$^{2+}$ binding markedly stabilizes MICU2 as demonstrated in a thermal stability assay (*Kamer et al., 2017*), allowing it to have a more stable interface for dimerization.

Major features of the uniplex structure help to reconcile some earlier experimental observations. First, initial studies showed that MCU exists in a large protein complex of ~500 kDa (*Baughman et al., 2011*; *Sancak et al., 2013*), which is compatible with the V-shaped MCU-EMRE subcomplex dimer bridged by a MICU1/MICU2 heterotetramer in the presence of Ca$^{2+}$ or blocked by two MICU1-MICU2 dimers in the absence of Ca$^{2+}$. Second, functional studies in knockout cell lines using the HEK293T cell system have demonstrated non-equivalence of MICU1 and MICU2 both at a genetic level and at a functional level (*Kamer and Mootha, 2014*). Specifically, the physical interaction between MCU and MICU2 requires the presence of MICU1, however, MCU and MICU1 were able to interact in the absence of MICU2. Genetic studies demonstrated that MICU1 is able to influence the threshold of calcium uptake by the uniporter independent of MICU2; however, the ability of MICU2 to do so required the presence of MICU1. The uniplex structure provides insight into this non-equivalence. Third, recent high-resolution structures of the MICU1-MICU2 heterodimer have revealed that MICU2 is capable of engaging in 'head-to-head' interactions as well as 'back-to-back' interactions (*Park et al., 2020*). This was puzzling but is now reconciled in the context of the MICU1-MICU2-MICU2-MICU1 bridge between the V-shaped MCU-EMRE dimer.

The MCU-EMRE subcomplex, which functions as a conductive channel, remains virtually identical in all our uniplex structures, indicating that the binding of MICU1-MICU2 does not introduce any conformational change to the channel. The uniporter gating mechanism is likely defined by how MICU1-MICU2 interacts with MCU-EMRE in a Ca$^{2+}$-dependent manner. In the presence of Ca$^{2+}$, the MICU1-MICU2 tetramer bridges the two MCU-EMRE subcomplexes without blocking the MCU channel pore; electrostatic interactions between MICU1 and EMRE mediate this unblocked uniplex assembly. In the absence of Ca$^{2+}$, MICU1 from the MICU1-MICU2 dimer engages in more extensive interactions with both EMRE and the MCU pore of the MCU-EMRE subcomplex, blocking the channel's external entrance. Because of the symmetry of the pore, the MICU1-MICU2 dimer can block the channel in four possible orientations. Interestingly, a portion of the apo MICU1-MICU2 can still dimerize to form a tetramer and bridge the MCU-EMRE subcomplex dimer without blocking the channel, similarly to the Ca$^{2+}$-bound uniplex. However, this bridging assembly in the apo state does not appear to be as tight as that in the Ca$^{2+}$-bound uniplex.

The conversion between the apo, blocked and the bridged, unblocked uniplex assemblies involves the formation of completely different inter-protein contacts and cannot be achieved by a simple conformational change. In order to transition from one state to another, MICU1-MICU2 has to first detach from the MCU-EMRE subcomplex and then re-attach at a different location. This leads to a simple working model for the gate-keeping mechanism of the mitochondrial calcium uniporter (*Figure 6*). The uniplex assembly with the apo MICU1-MICU2 dimer blocking the MCU pore prevails when cytosolic $Ca^{2+}$ is low, mitigating $Ca^{2+}$ uptake by the uniporter. Upon an elevation in cytosolic $Ca^{2+}$, binding of $Ca^{2+}$ to the EF-hands of MICU1-MICU2 triggers a conformational change at the MICU1-MICU2 dimerization interface and results in a downward movement of MICU2 that would directly interfere with MICU1's C-helix or the membrane surface. This steric interference prevents the dimer from maintaining its blocking interactions with the channel pore and causes the $Ca^{2+}$-bound MICU1-MICU2 to dissociate from and disinhibit MCU. The detached $Ca^{2+}$-bound MICU1-MICU2 further dimerizes and re-attaches to the MCU-EMRE subcomplex to form a bridged uniplex with the unblocked channel pore. The bridged uniplex assembly at high $Ca^{2+}$ keeps MICU1-MICU2

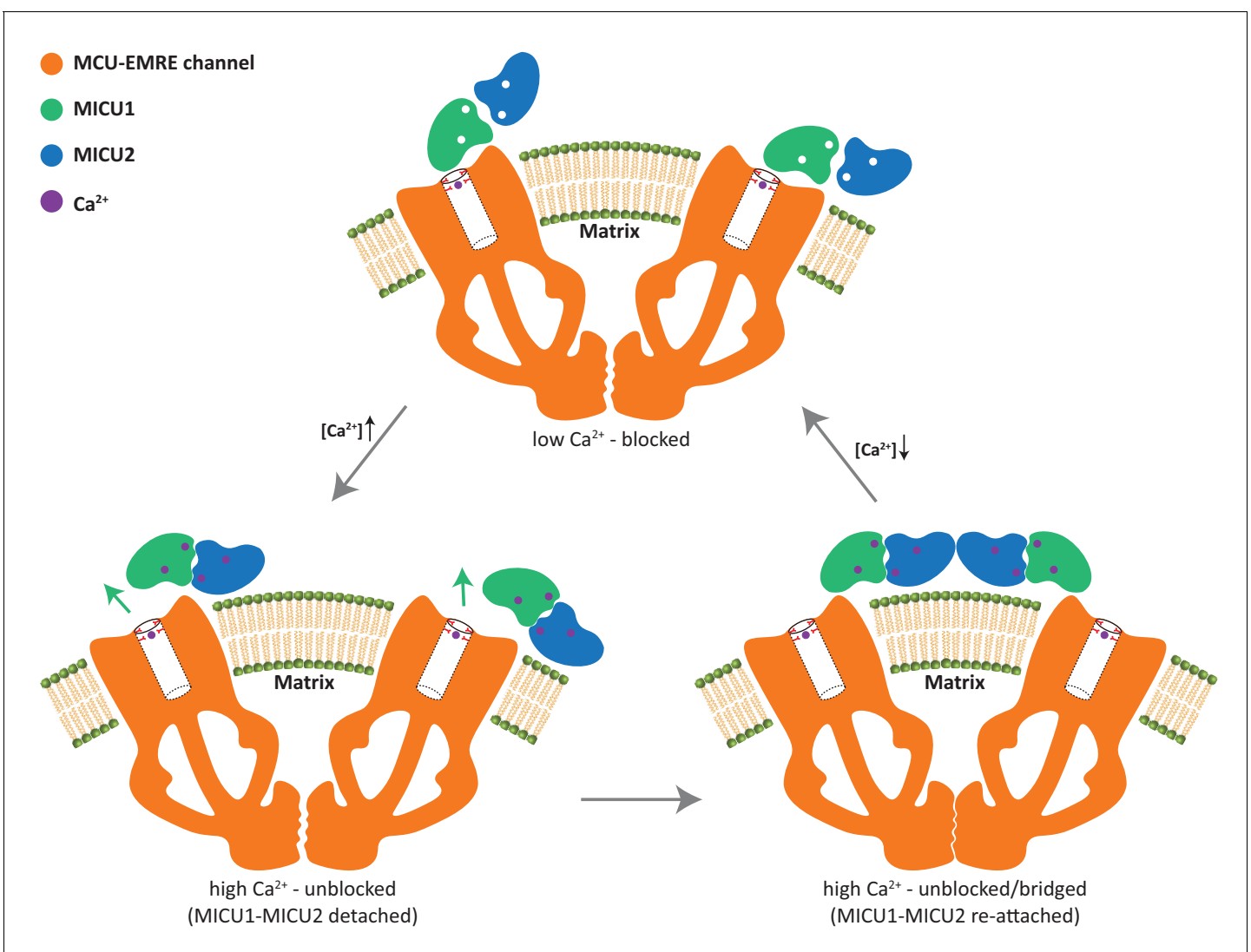

**Figure 6.** Working model for the gate-keeping mechanism of the mitochondrial calcium uniporter. The apo MICU1-MICU2 dimer binds and blocks the MCU pore when cytosolic $Ca^{2+}$ is low. Upon an elevation in cytosolic $Ca^{2+}$, $Ca^{2+}$ induced conformational changes within MICU1-MICU2 prevents the dimer from maintaining its blocking interactions with the channel pore and causes the $Ca^{2+}$-bound MICU1-MICU2 to dissociate from MCU. The detached MICU1-MICU2 dimerizes and re-attaches to the MCU-EMRE subcomplex to form a bridged uniplex with the unblocked channel pore.

in close proximity to the channel, making it readily available to block the MCU pore when $Ca^{2+}$ levels return to the resting concentration.

Among all those recently published uniplex structures (*Fan et al., 2020*; *Wang et al., 2020*; *Zhuo et al., 2020*), our results are consistent with the main findings from that of *Fan et al., 2020*, which showed a bridged, unblocked uniplex assembly at high $Ca^{2+}$ and a blocked assembly under low $Ca^{2+}$. A similar blocked uniplex structure was also observed in the study by *Wang et al., 2020*, even though the holocomplex is an artificial assembly between beetle MCU-EMRE and human MICU1-MICU2. The structure by *Zhuo et al., 2020*, which presumably represents the uniplex in the absence of $Ca^{2+}$, appears to resemble that of the apo, bridged uniplex in our study. Our structural observation that MICU1 exerts its gating effect on MCU by directly blocking the ion conduction pathway in the absence of $Ca^{2+}$ is in direct contradiction to the recent electrophysiological analysis of the uniporter (*Garg et al., 2020*), which argues that MICU1 does not block MCU at all. Given the current disagreement, more studies will be needed to reconcile these discrepancies.

# Materials and methods

## Key resources table

| Reagent type (species) or resource | Designation | Source or reference | Identifiers | Additional information |
|---|---|---|---|---|
| Gene (*Homo sapiens*) | *Hs*MCU | doi:10.1038/nature10234; doi:10.1038/nature10230 | NCBI:NM_001270679.1 | |
| Gene (*Homo sapiens*) | *Hs*EMRE | doi:10.1126/science.1242993 | NCBI:NM_033318.5 | |
| Gene (*Homo sapiens*) | *Hs*MICU1 | doi:10.1038/nature09358 | NCBI:NM_001195518.2 | |
| Gene (*Homo sapiens*) | *Hs*MICU2 | doi:10.1371/journal.pone.0055785 | NCBI:NM_152726.3 | |
| Recombinant DNA Reagent | pEZT-BM (plasmid) | doi:10.1016/j.str.2016.03.004 | Addgene 74099 | |
| Recombinant DNA Reagent | pET-28a (plasmid) | doi:10.1016/0022-2836(91)90856-2 | Novagen 69864–3 | |
| Strain, strain Background (*Escherichia coli*) | TOP10 | Thermo Fisher Scientific | Cat# 18258012 | Competent cells |
| Strain, strain Background (*Escherichia coli*) | BL21(DE3) | Thermo Fisher Scientific | Cat# EC0114 | Competent cells |
| Strain, strain Background (*Escherichia coli*) | DH10bac | Thermo Fisher Scientific | Cat# 10361012 | Competent cells |
| Cell line (*Spodoptera frugiperda*) | SF9 | Thermo Fisher Scientific | Cat# 11496015; RRID:CVCL_0549 | |
| Cell line (*Homo sapiens*) | HEK293F | Thermo Fisher Scientific | Cat# R79007; RRID:CVCL_D603 | |
| Software, algorithm | MotionCor2 | *Zheng et al., 2017* | http://msg.ucsf.edu/em/software/motioncor2.html | |
| Software, algorithm | GCTF | *Zhang, 2016* | https://www.mrclmb.cam.ac.uk/kzhang/Gctf | |
| Software, algorithm | RELION | *Scheres, 2020* | http://www2.mrclmb.cam.ac.uk/relion | |

*Continued on next page*

*Continued*

| Reagent type (species) or resource | Designation | Source or reference | Identifiers | Additional information |
|---|---|---|---|---|
| Software, algorithm | Chimera | *Pettersen et al., 2004* | https://www.cgl.ucsf.edu/chimera; RRID:SCR_004097 | |
| Software, algorithm | Pymol | Schrödinger | https://pymol.org/2; RRID:SCR_000305 | |
| Software, algorithm | COOT | *Emsley et al., 2010* | https://www2.mrclmb.cam.ac.uk/personal/pemsley/coot; RRID:SCR_014222 | |
| Software, algorithm | PHENIX | *Adams et al., 2010* | https://www.phenixonline.org | |
| Software, algorithm | OriginPro8 | OriginLab Corp. | https://www.originlab.com | |
| Others | QUANTIFOIL R1.2/1.3 | Quantifoil | | |

## Reconstitution of HsMCU-EMRE-MICU1-MICU2 holocomplex

Full-length human MICU1 (NCBI: NM_001195518.2) and MICU2 (NCBI: NM_152726.3) without mitochondrial targeting sequence (MTS) were PCR amplified from human cDNA library (purchased from the McDermott Center, UT Southwestern Medical Center) and sub-cloned into pET28a vectors with a C-terminal 8 × His tag using the restriction sites Xba I-Not I. *E. coli* (BL21 DE3) competent cells were freshly transformed, plated onto LB- kanamycin (LB-Kan) agar, and grown overnight. A single colony was picked from LB-Kan plate to seed LB media with 50 µg/mL kanamycin and grown in shaker flasks at 37°C in an orbital shaker until the cells reached an $OD_{600}$ = 1.0 ~ 1.2 at which point they were chilled on icy water for half an hour before protein expression was induced by addition of IPTG to a final concentration of 0.3 mM. The cells were then grown at 16°C for 20 hr before being harvested. In general, 6 L of *E. coli* culture were used for each purification of MICU1 or MICU2.

The same purification protocol was used for both MICU1 and MICU2. *E. coli* cell pellets from a 6-L culture were re-suspended in 100 mL of lysis buffer (20 mM Tris pH 8.0, 300 mM NaCl, 20 mM Imidazole, 2 mM DTT) and lysed by sonication followed by centrifugation at 40,000 x g for an hour. The supernatant was incubated with 3 ml Ni-NTA resins (QIAGEN) with gentle agitation at 4°C for an hour. The resin was then collected on a disposable gravity column and washed with 10 column volumes of lysis buffer. The bound protein was eluted with three column volumes of elution buffer (20 mM Tris pH 8.0, 300 mM NaCl, 500 mM Imidazole, 2 mM DTT). The protein eluent was concentrated to 500 µL using an Amicon centrifugal concentrator (50 kD cut-off, Millipore) and loaded onto a Superdex 200 10/300 GL column (GE Healthcare) pre-equilibrated with GF buffer A (20 mM Tris pH 8.0, 300 mM NaCl, 2 mM DTT) for further purification. Peak eluent containing MICU1 or MICU2 was collected for further reconstitution.

Human MCU-EMRE subcomplex was purified and reconstituted into nanodisc as was described previously (*Wang et al., 2019*). Briefly, Human MCU, EMRE, and MICU1 were co-expressed in HEK293f cells (Thermo Fisher Scientific) and purified in n-dodecyl-β-D-maltopyranoside detergent (DDM, Anatrace). MICU1 does not form a stable complex with MCU and EMRE but its co-expression is necessary for enhancing the expression of MCU and EMRE. Therefore, only the MCU-EMRE subcomplex was recovered after purification. The purified MCU-EMRE subcomplex in DDM was then reconstituted into nanodisc by incubating with Msp1 protein, lipids (POPC:POPE:POPG, 3:1:1 molar ratio) and BioBeads (BioRad) at 4°C. After incubation, the sample was further purified by gel filtration using a Superose 6 Increase 10/300 GL column (GE Healthcare) pre-equilibrated with GF buffer B (50 mM Tris pH 8.0, 50 mM NaCl). The main peak containing the MCU-EMRE subcomplex in nanodisc eluted around 13.6 mL.

The purified MICU1, MICU2 and HsMCU-EMRE in nanodisc were mixed with excess amounts of MICU1 and MICU2 based on SDS-PAGE estimation. The mixture was concentrated to 500 µl using an Amicon centrifugal concentrator (30 kD cut-off, Millipore) followed by dialysis against 1 L GF buffer C [50 mM Tris pH 8.0, 50 mM NaCl, 2 mM DTT, and 2 mM EGTA (for the $Ca^{2+}$-free state) or 2

mM $CaCl_2$ (for $Ca^{2+}$-bound state)] at 4˚C overnight. The sample after dialysis was further purified by Superose 6 Increase 10/300 GL column (GE Healthcare) pre-equilibrated with GF buffer C. The MCU-EMRE-MICU1-MICU2 holocomplex in nanodisc eluted around 12.5 mL and the protein components of the purified holocomplex were verified by SDS-PAGE analysis. The purified uniplex sample was immediately concentrated to 1.6 mg/mL for cryo-EM grid preparation.

Cell lines used in this study were purchased from and verified by Thermo Fisher Scientific. They were tested negative for mycoplasma contamination.

## EM data acquisition

The cryo-EM grids were prepared by applying 3 µL of the human MCU-EMRE-MICU1-MICU2 complex in nanodisc ($Ca^{2+}$-free or $Ca^{2+}$-bound state) to a glow-discharged Quantifoil R1.2–1.3 300-mesh gold holey carbon grid (Quantifoil, Micro Tools GmbH, Germany) and blotted for 4.0 s under 100% humidity at 4˚C before being plunged into liquid ethane using a Mark IV Vitrobot (FEI). Micrographs were acquired on a Titan Krios microscope (FEI) operated at 300 kV with a K3 Summit direct electron detector (Gatan), using a slit width of 20 eV on a GIF-Quantum energy filter. Serial EM (*Mastronarde, 2005*) software was used for automated data collection. A calibrated magnification of 105,000 × was used for imaging, yielding a pixel size of 0.833 Å on images. The defocus range was set at $-0.9 \sim -2.2$ µm. Each micrograph was dose-fractionated to 40 frames under a dose rate of 21 e⁻/pixel/s, with a total exposure time of 2 s, resulting in a total dose of about 60.5 e⁻/Å$^2$.

## Image processing

All movie frames were aligned using MotionCor2 (*Zheng et al., 2017*). CTF (Contrast transfer function) of motion-corrected micrographs was estimated using Gctf (*Zhang, 2016*). Data were processed using Relion 3.0.6 (*Zivanov et al., 2018*) for the $Ca^{2+}$-bound state, and Relion 3.1-beta (*Scheres, 2020*; *Zivanov et al., 2020*) for the $Ca^{2+}$-free state. All resolutions were reported according to the gold-standard Fourier shell correlation (FSC) using the 0.143 criterion (*Henderson et al., 2012*). Local resolution was estimated using Relion.

For the complex in the $Ca^{2+}$-bound state, a total of 4116 movies stacks were collected. Motion-corrected micrographs were then subjected to manual inspection after CTF estimation and 595 bad images were discarded, resulting in 3521 good micrographs. We first manually picked ~1000 particles in Relion and performed a 2D classification. The 2D class averages corresponding to the complex were then selected and used as templates for Auto-pick in Relion. A total number of 694,082 particles were auto-picked and subjected to one round of 2D classification. We selected 277,816 particles from this 2D classification and performed the initial round of 3D classification with particle aligned into six classes. The HsMCU-EMRE subcomplex map (*Wang et al., 2019*) low-pass filtered to 60 Å was used as the initial model. The second class with 59,406 particles showing clear features of MICU1-MICU2 heterotetramer binding to the HsMCU-EMRE subcomplex was chosen and refined with C2 symmetry. We then subtracted the density corresponding to the dimer of the channel and performed another round of 3D classification with the remaining heterotetrameric density into six classes. Two of the six classes (total 19,924 particles) showed MICU1-MICU2 density with good quality and were chosen for subsequent processing. We first refined this set of particles following CTF refinement in Relion and obtained a density map with an overall resolution of 4.17 Å (FSC = 0.143).

For the complex in the $Ca^{2+}$-free state, a total number of 15,093 movies were collected, from which we manually discarded 2078 bad micrographs after motion correction and CTF estimation. Using the template obtained from the high-$Ca^{2+}$ dataset, we auto-picked 2,070,698 particles from 13,015 good micrographs in Relion. After 2D classification, we selected 1,294,333 particles, which were then subjected to 3D classification using the map obtained from the complex in the $Ca^{2+}$-bound state and low-pass filtered to 60 Å as the initial reference. From this we observed one class of blocked uniplex with monomeric MCU-EMRE subcomplex (with 328,588 particles) and three classes of uniplex with dimeric MCU-EMRE subcomplex: competing (with 134,975 particles), blocking (with 120,792 particles), and bridging (with 160,962 particles). Those particles with the dimeric MCU-EMRE subcomplex were further classified with a mask around the MICU1-MICU2 region, and the best resolving subclasses were refined. We also classified the bridging class based on the density of the MCU-EMRE subcomplex and obtained a reconstruction with an overall resolution of 3.7 Å, indicating that the data are of good quality and the structure of the MCU-EMRE part of the uniplex is

well resolved. To better resolve the blocking conformation, particles with the monomeric MCU-EMRE subcomplex were re-selected from 2D classes and combined with the particles of the blocked uniplex with the dimeric MCU-EMRE subcomplex for the final structure determination as described below.

For the blocked uniplex with the dimeric MCU-EMRE subcomplex, we selected 69,343 particles from the second round of 3D classification. We then performed C2 symmetry expansion and subtracted one complete channel from this symmetry expanded particle stack. Next, we kept only the symmetrical region of the channel together with the bound MICU1-MICU2 density by another signal subtraction procedure. The resulting particles were then classified with a mask around the MICU1-MICU2 density. We observed that MICU1-MICU2 in this blocking conformation adopts four possible orientations that are 90 degrees apart, in accordance with the tetrameric nature of the MCU channel. We then manually rotated (by manipulating the AngleRot column in the star files) the other three classes to the same orientation as the first, which we defined as the consensus orientation, and combined them into one particle stack. Next, 2D classes with clear features of a monomeric MCU-EMRE subcomplex were re-selected. After two rounds of 3D classification, we obtained a subset of 126,457 particles, which were further classified based on the MICU1-MICU2 density. Next, signal subtraction was performed to remove the N-terminal domain of MCU, and the resulting particles were manually rotated to the consensus orientation and combined with the aforementioned particle stack from the dimeric channel followed by removal of duplicates. The resulting 171,471 particles were then classified with a mask around the MICU1-MICU2 region, and we obtained a class of 44,681 particles with the best MICU1-MICU2 density. This class was then auto-refined in Relion, resulting in a density map with an overall resolution of 4.6 Å.

## Model building, refinement, and validation

All EM maps of the HsMCU-EMRE-MICU1-MICU2 holocomplex show high quality density at the MCU-EMRE region, allowing us to confidently fit the structure of the MCU-EMRE subcomplex (*Wang et al., 2019*) (PDB: 6O5B) into the density using UCSF-Chimera. For the MICU1-MICU2 part of the uniplex in the $Ca^{2+}$-bound state, the models of human MICU1 (*Wang et al., 2014*) (PDB: 4NSD) and human MICU2 (*Wu et al., 2019*) (PDB: 6IIH), both in the $Ca^{2+}$-bound state, were used to fit into the density manually in Coot (*Emsley et al., 2010*). The regions that did not fit to the density map were adjusted as rigid bodies. The structures of the apo human MICU1 (*Wang et al., 2014*) (PDB: 4NSC) and apo human MICU2 (*Xing et al., 2019*) (PDB: 6AGJ) were used to model MICU1-MICU2 in the blocked uniplex structure in the $Ca^{2+}$-free condition. For the modeling of the MICU1-MICU2 regions in the low-resolution structures of the apo uniplex with bridging and competing configurations, the MICU1-MICU2 structure obtained from the $Ca^{2+}$-bound uniplex appeared to fit the density maps much better than that from the apo, blocked uniplex.

The structures of the $Ca^{2+}$-bound, unblocked uniplex and the apo, blocked uniplex were refined against summed maps using phenix.real_space_refine, with secondary structure restraints applied (*Adams et al., 2010*). Due to their resolution limit, the structures of the apo uniplex in the bridging and competing configurations were not refined. All the figures were prepared using PyMol (*Schrodinger LLC, 2015*), UCSF-Chimera (*Goddard et al., 2007*; *Pettersen et al., 2004*) and UCSF ChimeraX (*Goddard et al., 2018*).

## Acknowledgements

Single particle cryo-EM data were collected at the University of Texas Southwestern Medical Center Cryo-EM Facility that is funded by the CPRIT Core Facility Support Award RP170644. This work was supported in part by the Howard Hughes Medical Institute (YJ and VKM) and by grants from the National Institute of Health (GM079179 to Y J and R01GM136976 to X-CB), the Welch Foundation (Grant I-1578 to Y J and I-1944 to X-CB), the Cancer Prevention and Research Institute of Texas (RP160082 to X-CB), and the Virginia Murchison Linthicum Scholar in Medical Research fund (to X-CB).

## Additional information

### Funding

| Funder | Grant reference number | Author |
|---|---|---|
| Howard Hughes Medical Institute | | Vamsi K Mootha<br>Youxing Jiang |
| National Institute of General Medical Sciences | GM079179 | Youxing Jiang |
| National Institute of General Medical Sciences | GM136976 | Xiao-chen Bai |
| Welch Foundation | I-1578 | Youxing Jiang |
| Welch Foundation | I-1944 | Xiao-chen Bai |
| Cancer Prevention and Research Institute of Texas | RP160082 | Xiao-chen Bai |
| Virginia Murchison Linthicum Scholar in Medical Research | | Xiao-chen Bai |

The funders had no role in study design, data collection and interpretation, or the decision to submit the work for publication.

### Author contributions

Yan Wang, Conceptualization, Data curation, Formal analysis, Validation, Investigation, Visualization, Methodology, Writing - original draft, Writing - review and editing; Yan Han, Data curation, Formal analysis, Validation, Investigation, Visualization, Methodology, Writing - original draft, Writing - review and editing; Ji She, Data curation, Methodology; Nam X Nguyen, Investigation, Writing - review and editing; Vamsi K Mootha, Conceptualization, Investigation, Writing - review and editing; Xiao-chen Bai, Data curation, Investigation, Methodology, Writing - review and editing; Youxing Jiang, Conceptualization, Supervision, Funding acquisition, Validation, Investigation, Writing - original draft, Project administration, Writing - review and editing

### Author ORCIDs

Yan Wang (iD) https://orcid.org/0000-0001-8243-693X
Yan Han (iD) https://orcid.org/0000-0002-1207-7756
Ji She (iD) http://orcid.org/0000-0001-7006-6230
Vamsi K Mootha (iD) http://orcid.org/0000-0001-9924-642X
Xiao-chen Bai (iD) http://orcid.org/0000-0002-4234-5686
Youxing Jiang (iD) https://orcid.org/0000-0002-1874-0504

### Decision letter and Author response

Decision letter https://doi.org/10.7554/eLife.60513.sa1
Author response https://doi.org/10.7554/eLife.60513.sa2

## Additional files

### Supplementary files
• Transparent reporting form

### Data availability

The cryo-EM density maps of the human MCU-EMRE-MICU1-MICU2 holocomplex have been deposited in the Electron Microscopy Data Bank under accession numbers EMD-22215 for the Ca2+-bound state, EMD- 22216 for the apo, blocked state, EMD-22213 for the apo, bridging state and EMD-22214 for the apo, competing state. Atomic coordinates have been deposited in the Protein

Data Bank under accession numbers 6XJV for the Ca2+-bound state and 6XJX for the apo, blocked state.

The following datasets were generated:

| Author(s) | Year | Dataset title | Dataset URL | Database and Identifier |
|---|---|---|---|---|
| Wang Y, Jiang Y | 2020 | human MCU-EMRE-MICU1-MICU2 holocomplex, Ca2+-bound state | http://www.ebi.ac.uk/pdbe/entry/emdb/EMD-22215 | Electron Microscopy Data Bank, EMD-22215 |
| Wang Y, Jiang Y | 2020 | human MCU-EMRE-MICU1-MICU2 holocomplex, apo, blocked state | http://www.ebi.ac.uk/pdbe/entry/emdb/EMD-22216 | Electron Microscopy Data Bank, EMD-22216 |
| Wang Y, Jiang Y | 2020 | human MCU-EMRE-MICU1-MICU2 holocomplex, apo, bridging state | http://www.ebi.ac.uk/pdbe/entry/emdb/EMD-22213 | Electron Microscopy Data Bank, EMD-22213 |
| Wang Y, Jiang Y | 2020 | human MCU-EMRE-MICU1-MICU2 holocomplex, apo, competing state | http://www.ebi.ac.uk/pdbe/entry/emdb/EMD-22214 | Electron Microscopy Data Bank, EMD-22214 |
| Wang Y, Jiang Y | 2020 | Atomic coordinates for the Ca2+-bound state | http://www.rcsb.org/structure/6XJV | RCSB Protein Data Bank, 6XJV |
| Wang Y, Jiang Y | 2020 | Atomic coordinates for the apo, blocked state | http://www.rcsb.org/structure/6XJX | RCSB Protein Data Bank, 6XJX |

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
