## [Decision Letter]

**Acceptance summary:**

This manuscript from the Jiang lab presents cryo EM structures of the human mitochondrial calcium uniporter (MCU) holocomplex that senses cytoplasmic calcium and mediates calcium uptake into mitochondria. When calcium is present, the MICU1/MICU2 heterodimer from one holocomplex swings away from the pore and interacts with the MICU1/MICU2 heterodimer of the second complex, allowing calcium to access the ion permeation pore. When calcium is chelated with EGTA, the particles are more heterogenous, with the most highly represented state showing MICU1 interacting with the MCU pore, presumably blocking calcium conduction. These findings are important because they provide critical information on the mechanism by which calcium ions regulate MCU to open a permeation pathway for calcium to enter mitochondria. The work is of high quality, expands and strongly enhances the results of three other groups who have also recently reported structures of this important complex. The authors have done a nice job of revising the manuscript to incorporate the reviewer's suggestions. This manuscript represents an important contribution to a highly competitive field.

**Decision letter after peer review:**

Thank you for submitting your article "Structural insights into the Ca^2+^-dependent gating of the human mitochondrial calcium" for consideration by *eLife*. Your article has been reviewed by three peer reviewers, and the evaluation has been overseen by Kenton Swartz as the Reviewing and Senior Editor. The following individual involved in review of your submission has agreed to reveal their identity: Alexander Sobolevsky (Reviewer #2).

The reviewers have discussed the reviews with one another and the Reviewing Editor has drafted this decision to help you prepare a revised submission.

Summary:

This manuscript from the Jiang lab presents cryo EM structures of human MCU in the presence and absence of Ca^2+^. When Ca^2+^ is present, the MICU1/MICU2 heterodimer from one holocomplex swings away from the pore and interacts with the MICU1/MICU2 heterodimer of the second complex, allowing Ca^2+^ to access the pore. When EGTA is present, and Ca^2+^ presumably absent, the particles are more heterogenous, with the most highly represented state showing MICU1 interacting with the MCU pore, presumably blocking Ca^2+^ conduction. The structures presented reprise previous structures by other labs, and the present structures are overall in accord with these. We realize that this is a highly competitive and important area of investigation, but we think the present manuscript would be strengthened by addressing the following concerns.

Essential revisions:

1) The manuscript is written under the assumption that it presents the first structure of the holocomplex. Since two other groups have recently published the structures of the holocomplex for human (Fan et al., 2020; Zhuo et al., 2020) and one group for beetle (Wang et al., 2020), the authors should not pretend that their structure is the first. To make this manuscript most useful for the reader, we request that you mention all these publications in the beginning of the manuscript, not at the end, and enrich this manuscript with comprehensive comparisons across all reported structures throughout the text.

2) Several types of contacts between MICU1 and MCU-EMRE are described in Figure 3. It would be nice to have an assessment of physiological significance at least for some of these interactions through mutagenesis and functional recordings. This seems especially appropriate given the recent electrophysiological analysis mentioned by the authors (Garg et al., 2020), which argues that MICU1 does not block MCU at all.

3) MICU1 and MICU2 are very similar in terms of sequence and structure, and known to homo-dimerize. As the authors point out, structures are available of the homodimers. The protein purification protocol, which involves mixing MICU1 and MICU2 with MCU/EMRE in nanodiscs, does not preclude a heterogenous mix of heterodimeric MICUs and homodimeric MICUs. Did the authors perform any biochemical controls to ensure that complexes contained heterodimers and not homodimers?

4) Given that the maps are not great in the region of MICU1/MICU2 for the various EGTA structures, could MICU1/MICU1 homodimers be docked into the MICU density of the maps for any of these structures? Could this contribute to the conformational heterogeneity?

5) Although the authors identify a number of residues from MICU1 and MICU2 involved in blocking the MCU pore, there is not very reliable sidechain density in the blocked structures. The authors do acknowledge this in the text, but also show sidechains in Figure 3. Can the authors show the maps in some of these close-up views of the interaction between MCU and MICU1? Experimental data to probe whether some of the residues that they identify would augment their argument about sidechains important to blocking interactions.

6) In the Materials and methods, the authors mention that the bridging and competing states from the EGTA condition are fit better by the Ca^2+^ bound forms. Could the authors expand on this in the main text? Do the authors think this a consequence of incomplete Ca^2+^ chelation, or an ensemble of calcium-bound MICUs?

---

## [Author Response]

Essential revisions:1) The manuscript is written under the assumption that it presents the first structure of the holocomplex. Since two other groups have recently published the structures of the holocomplex for human (Fan et al., 2020; Zhuo et al., 2020) and one group for beetle (Wang et al., 2020), the authors should not pretend that their structure is the first. To make this manuscript most useful for the reader, we request that you mention all these publications in the beginning of the manuscript, not at the end, and enrich this manuscript with comprehensive comparisons across all reported structures throughout the text.

As suggested, in the revised manuscript we have mentioned the structural studies of other groups in the Introduction section and also discussed the similarities and differences among these structures to our study where appropriate in the Results and Discussion sections.

2) Several types of contacts between MICU1 and MCU-EMRE are described in Figure 3. It would be nice to have an assessment of physiological significance at least for some of these interactions through mutagenesis and functional recordings. This seems especially appropriate given the recent electrophysiological analysis mentioned by the authors (Garg et al., 2020), which argues that MICU1 does not block MCU at all.

Mutagenesis studies related to the three regions in MICU1 that participate in blocking interactions have been performed before, including the two recent structural studies (Fan et al., 2020 and Wang et al., 2020). These functional studies have been clearly mentioned and referenced in the revision. The key blocking interactions suggested from our structure are consistent with the structural studies of the apo holocomplex from both Feng’s group (Fan et al., 2020) and Long’s group (Wang et al., 2020). This point has been clearly stated in the revision.

3) MICU1 and MICU2 are very similar in terms of sequence and structure, and known to homo-dimerize. As the authors point out, structures are available of the homodimers. The protein purification protocol, which involves mixing MICU1 and MICU2 with MCU/EMRE in nanodiscs, does not preclude a heterogenous mix of heterodimeric MICUs and homodimeric MICUs. Did the authors perform any biochemical controls to ensure that complexes contained heterodimers and not homodimers?

Two experimental observations in our study suggest that MICU1 and MICU2 do form a heterodimer in the majority of the holocomplex. (1) SDS page of the purified complex (Figure 1—figure supplement 2 in revised manuscript) shows equivalent amount of MICU1 and MICU2. Since MICU2 by itself does not interact with MCU-EMRE, it has to form a heterodimer with MICU1 in order to associate with the complex. (2) There is a clear structural difference at the N-terminal region between MICU1 and MICU2, allowing us to distinguish MICU1 from MICU2 in the EM maps.

4) Given that the maps are not great in the region of MICU1/MICU2 for the various EGTA structures, could MICU1/MICU1 homodimers be docked into the MICU density of the maps for any of these structures? Could this contribute to the conformational heterogeneity?

The MICU1/MICU1 homodimer does not dock into the maps of the complex with EGTA as well as MICU1/MICU2 heterodimer. As mentioned in our response to previous point, the majority (if not all) of the holocomplex should contain the MICU1/MICU2 heterodimer. We do not expect that the formation of MICU1/MICU1 homodimer, if any, is the cause of the conformational heterogeneity.

5) Although the authors identify a number of residues from MICU1 and MICU2 involved in blocking the MCU pore, there is not very reliable sidechain density in the blocked structures. The authors do acknowledge this in the text, but also show sidechains in Figure 3. Can the authors show the maps in some of these close-up views of the interaction between MCU and MICU1? Experimental data to probe whether some of the residues that they identify would augment their argument about sidechains important to blocking interactions.

As we stated in the manuscript, we cannot see the sidechain density of most residues due to the resolution limit of our structures. However, the main-chain positions can be reasonably well defined with the help of known MICU1 and MCU-EMRE structures. The protein/protein interactions in the blocking state are inferred based on the close proximity of those charged or hydrophobic residues. We feel it is necessary to model the side-chains of these residues instead of using alanines in order to provide a clear picture of the locations of these residues. As mentioned in our previous response, similar blocking interactions have also been observed in the two recent structures of the apo complex (Fan et al., 2020 and Wang et al., 2020). Mutagenesis of some key residues have also been tested in their studies. In the revised manuscript, we modified the description of these putative interactions to avoid any over interpretation of our structure.

6) In the Materials and methods, the authors mention that the bridging and competing states from the EGTA condition are fit better by the Ca^2+^ bound forms. Could the authors expand on this in the main text? Do the authors think this a consequence of incomplete Ca^2+^ chelation, or an ensemble of calcium-bound MICUs?

We have expanded our discussion on the bridging and competing forms of the apo complex (in the presence of EGTA) in the main text as suggested. In the presence of 2 mM EGTA, the free [Ca^2+^] is extremely low (~ 1.72e-11 M) and we don’t believe the formation of the bridged apo complex is caused by incomplete Ca^2+^ chelation. As we mentioned in the Discussion, our apo bridged holocomplex is very similar to the structure by Zhuo et al. (Zhuo et al., 2020), which is also obtained in the absence of Ca^2+^.